# CPI203, a BET inhibitor, down-regulates a consistent set of DNA synthesis genes across a wide array of glioblastoma lines

Matthew C. Garrett[1#‡*], Troy Carnwath[2#], Rebecca Albano[3], Yonghua Zhuang[4], Catherine A. Behrmann[5], Merissa Pemberton[5], Farah Barakat[6], Robert Lober[7], Mark Hoeprich[1], Anthony Paravati[8], Marilyn Reed[1], Hailey Spry[1], Daniel Woo[9], Eric O'Brien[10], Brett VanCauwenbergh[10], John Perentesis[10], Rani Nasser[11], Mario Medvedovic[12], David R. Plas[5‡]

1 Department of Neurosurgery, Kettering Health Network, Kettering, Ohio, United States of America, 2 University of Cincinnati College of Medicine, Cincinnati, Ohio, United States of America, 3 Department of Neurosurgery, University of Cincinnati College of Medicine, Cincinnati, Ohio, United States of America, 4 Biostatistics and Bioinformatics Shared Resource, University of Colorado Cancer Center, University of Colorado Anschutz Medical Campus, Aurora, Colorado, United States of America, 5 Department of Cancer Biology, University of Cincinnati, Cincinnati, Ohio, United States of America, 6 Alfaisal University, College of Medicine, Riyadh, Saudi Arabia, 7 Division of Neurosurgery, Dayton Children's Hospital, Dayton, Ohio, United States of America, 8 Department of Radiation Oncology, Kettering Health Network, Kettering, Ohio, United States of America, 9 Department of Neurology, University of Cincinnati College of Medicine, Cincinnati, Ohio, United States of America, 10 Division of Experimental Hematology and Cancer Biology, Cincinnati Children's Hospital Medical Center, Department of Pediatrics, University of Cincinnati College of Medicine, Cincinnati, Ohio, United States of America, 11 Department of Neurosurgery, University of Cincinnati College of Medicine, Cincinnati, Ohio, United States of America, 12 Department of Environmental and Public Health Sciences, University of Cincinnati College of Medicine, Cincinnati, Ohio, United States of America

#Both authors contributed equally
‡ Co-communicating authors
* matthew.garrett2@ketteringhealth.org

## Abstract

### Introduction

Glioblastomas utilize malignant gene expression pathways to drive growth. Many of these gene pathways are not directly accessible with molecularly targeted pharmacological agents. Chromatin-modifying compounds can alter gene expression and target glioblastoma growth pathways. In this study, we utilize a systematic screen of chromatin-modifying compounds on a panel of patient-derived glioblastoma lines to identify promising compounds and their associated gene targets.

### Methods

Five glioblastoma cell lines were subjected to a drug screen of 106 chromatin-modifying compounds representing 36 unique drug classes to determine the twelve most promising drug classes and the best candidate inhibitors in each class. These twelve drugs were then tested with a panel of twelve patient-derived gliomasphere

**Data availability statement:** The datasets generated and/or analyzed during the current study are available in the Garrett Lab Data Directory (https://github.com/TroyCarnwath/garrett_lab).

**Funding:** MG was supported by University of Cincinnati Dean's Funds and the 2021-2022 B*CURED and NREF Young Clinician Investigator Award. RA was supported by a T32 CA236764-04 Training Grant. TC was supported by an NREF Medical Summer Research Fellowship. The funders had no role in study design, data collection, analysis, decision to publish or preparation of the manuscript.

**Competing interests:** No authors have competing interests.

lines to identify growth inhibition and corresponding gene expression patterns. Overlap analysis and weighted co-expression network analysis (WCGNA) were utilized to determine potential target genes and gene pathways.

## Results

The initial drug screen identified twelve candidate pharmacologic agents for further testing. Drug sensitivity testing indicated an overall high degree of variability between gliomasphere lines. However, CPI203 was the most consistently effective compound, and the BET inhibitor class was the most consistently effective class of compounds across the gliomasphere panel. Correspondingly, most of the compounds tested had highly variable effects on gene expression between gliomasphere lines. CPI203 stood out as the only compound to induce a consistent effect on gene expression across different gliomasphere lines, specifically down-regulation of DNA-synthesis genes. Amongst the twelve tested cell lines, high expression of CDKN2A and CDKN2B distinguished more drug sensitive from more drug resistant lines. WCGNA identified two oncogenic gene modules (FBXO5 and MELK) that were effectively downregulated by CPI203 (FBXO5) and ML228 (FBXO5 and MELK).

## Conclusions

The bromodomain inhibitor CPI203 induced relatively consistent effects on gene expression and growth across a variety of glioblastoma lines, specifically down-regulating genes associated with DNA replication. We propose that clinically effective BET inhibitors have the potential to induce consistent beneficial effects across a spectrum of glioblastomas.

---

## Introduction

Glioblastoma is the most common primary brain tumor and despite decades of research, the prognosis remains poor and effective treatment options are limited [1]. Amplifications of pro-growth genes (EGFR) as well as deletions of tumor suppressor genes (PTEN, p53, Rb) are frequently occurring driver events in glioblastoma, but therapeutics designed to counteract pathways regulated by these oncogenic drivers have not established a survival benefit in conventionally designed cohort-based clinical trials [2–4]. In addition to pathologic alterations in the base pair DNA sequences (i.e., genetics), the regulation of gene expression by DNA or histone modifications (i.e., epigenetics) plays a key role in glioblastoma growth and progression. The cell has multiple mechanisms to regulate gene transcription. Glioblastomas can often hijack these mechanisms to alter DNA repair capacity(e.g., MGMT [4,5]) or prevent senescence (e.g., TERT [6]). Genetic-epigenetic interactions are prominent in gliomas – a single base-pair mutation in the active site of the IDH metabolic enzyme results in DNA [7] and histone [8] hyper-methylation and may be the initial mutation that leads to the creation of low-grade gliomas [9]. Similarly, a large proportion

of diffuse intrinsic pontine gliomas (DIPGs) harbor a mutation in the H3 gene (H3K23M [10]) leading to the opening of histone architecture and increased gene expression [11]. While it is not possible to pharmacologically replace deleted genomic DNA, it may be possible to pharmacologically reprogram the chromatin structure and restore normal gene expression programs. Many such chromatin-modifying compounds are under investigation (HDAC inhibitors [12–16], BET inhibitors [17,18], Decitabine [19–22]).

Unfortunately, despite decades of research, there have been relatively few interventions shown to have any positive effect on overall survival in glioblastoma [1,23,24]. One reason for this may be the methods by which candidate compounds are tested in the preclinical and clinical setting. Glioblastoma is a highly heterogenous disease both between patients and even within a single tumor. Preclinical results may be highly cell and context dependent and will not be reproduced when tested against a highly heterogenous patient population. Fortunately, recent evidence from Ranjan et al. suggests that patient-derived low passage 3-dimensional gliomasphere lines capture many of the aspects of their parent tumors, including their drug sensitivity profile [25]. Expanding this principle, it may be informative to test potential clinical trial drugs on large panels of gliomasphere lines to eliminate those without robust and consistent efficacy. To that end, the present study integrates gliomasphere drug efficacy data with transcriptional responses to therapeutics to identify chromatin-modifying compounds with consistent and robust effects across a variety of heterogenous glioblastoma cell lines.

## Results

### Different glioblastoma cell lines have distinct drug sensitivity profiles. CPI203 and BET inhibitors are the most consistently effective drug and drug class respectively

A drug screen of 106 chromatin modifying/epigenetic compounds targeting 36 different genes was conducted in five independently derived gliomasphere lines(Supplementary Figure 1). Gliomaspheres were plated at optimum conditions to prevent over-growth during the time tested. Cell viabilities were measured via an MTS assay, and "Area under the Curve" (AUC) analysis was calculated at three days (1uM and 10uM) and four weeks(500nM) in duplicate. Fresh media, growth factors and drug were supplemented weekly. Considering that compounds exerting their main effects through epigenetic reprogramming would require a longer timeframe [7,26] and wanting to eliminate acute non-specific cyto-toxicity we computed a "delta AUC" by subtracting the MTS AUC at four weeks from the AUC at three days. An example of this calculation for three candidate compounds (rucaparib, decitabine and CPI203) is shown in Fig 1A. A heatmap of these delta UACs for all compounds tested is shown in Fig 1B. Each compound was grouped according to its gene target. The list of gene targets and the number of compounds in each group is listed (Fig 1C). The compounds were ranked by delta AUC and an average rank for each group was calculated by averaging the rank of each compound in that group. BRPF was the least effective compound class(lowest average rank) and was used as a reference to calculate p-values by t-test. CPI203 specifically (Fig 1D) and BET inhibitors more generally (Fig 1E) were found to be the most consistently effective inhibitor and inhibitor class respectively. The top twelve compound classes/groups were ranked according to average AUC score. The compound with the highest delta AUC within each of those twelve groups was selected for further testing (Fig 1E). A volcano plot was constructed by computing the delta AUC (x-axis) and p-value (y-axis) between the viabilities at day 3 (1uM and 10uM) and week 4 (500nM) (Fig 1F).

### Chromatin-modifying compounds can act as gene up-regulators or gene down-regulators

To further expand the analysis of variability seen in the initial drug screen, an additional twelve gliomasphere lines were exposed to twelve different inhibitors for seven days. Viability was tested at four different dosages (10uM, 1uM, 100nM and 10nM) and run in triplicate. An IC-50 was calculated using logistic regression (Fig 2A–2B) using GraphPad version 10.1.1 for Mac OS (www.graphpad.com). There was again a high degree of heterogeneity in sensitivity between the cell lines and many of the compounds showed a variability in IC-50 of over 1000-fold between sensitive and resistant lines (e.g., hesperidin IC-50 range: 3nM-100uM) while other compounds were more consistent

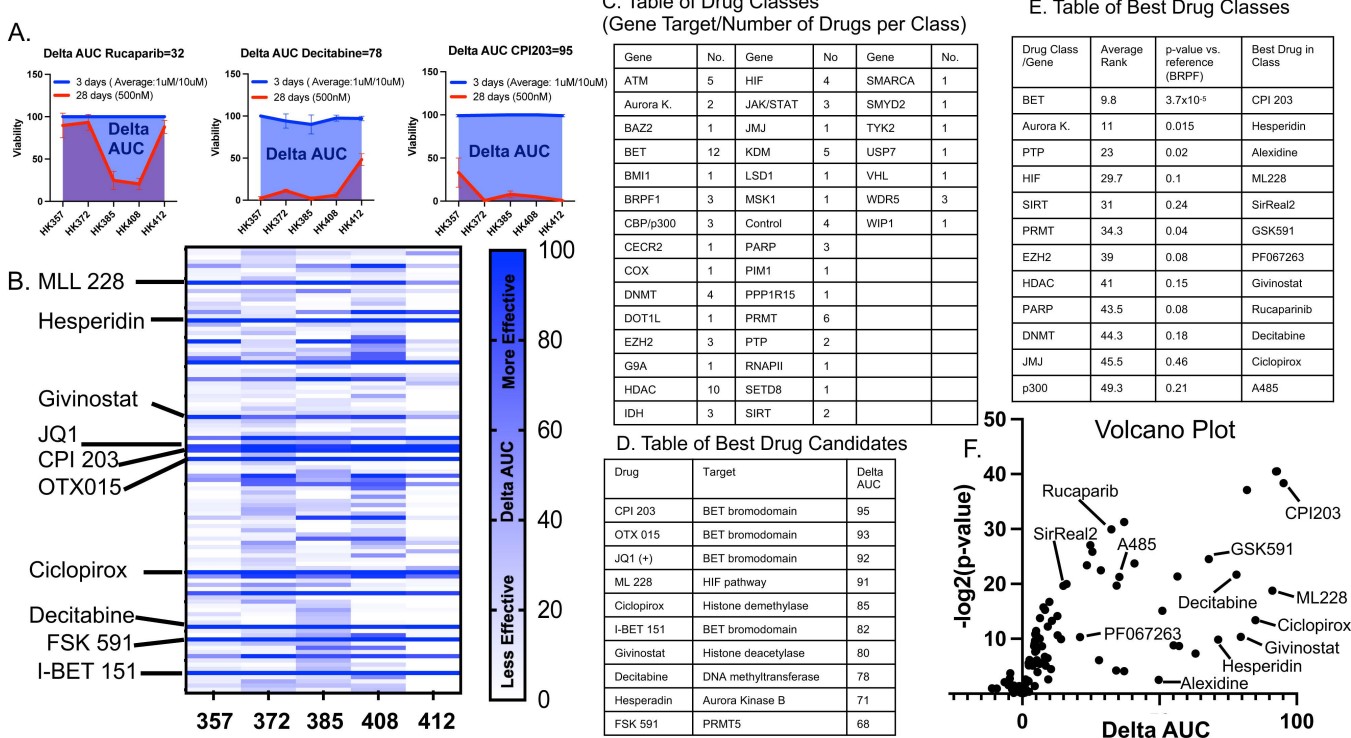

**Fig 1. 106 compounds targeting 35 different genes were tested on five glioblastoma primary cultures (357, 372, 385, 408, 412) and tested for viability at day 3 (1um and 10uM) and day 28 (500nM) with growth factor and drug replenished weekly.** In order to identify compounds that would slow growth over time, a delta AUC was calculated (delta AUC = AUC 3days-AUC 28days). A. An example of this calculation for three compounds (rucapraib, decitabine and CPI203) is shown. B. The delta AUCs for each compound was plotted as a heatmap with the ten best candidate compounds shown. C. Each compound was categorized based on the target gene of inhibition. The gene targets and the number of drugs per group is shown. D. The ten best compounds were ranked by highest delta AUC. E. All 106 compounds were ranked by delta AUC. An average rank for each group was calculated by averaging the ranks of all the compounds in that group. A p-value was calculated (t-test) by comparison to BRPF (the lowest group). F. Viabilities at 3 days (1uM and 10uM) and 28 days (500nM) were calculated for each compound and plotted as a volcano plot y-axis (p-value of viabilities at 3day vs. 28day t-test) and x-axis (delta-AUC).

(ML228 IC-50 range: 0.1uM-5uM) (Fig 2A–2B). The mean and standard deviation of the IC-50s for each compound are shown in Fig 2A. Interestingly, the gliomasphere line HK-385 showed a high level of resistance to all the tested compounds.

Gene expression analysis (RNA-seq) was used to determine the gene targets of these compounds. To best determine genetic effects without non-specific toxicity, we identified the drug dosage that yielded an approximate 65% viability after a 7-day treatment period (Supplemental Figure 2). RNA-seq was used to compare drug-treated and control samples to identify gene targets. The number of genes with ≥ 2-fold change from control were tabulated and shown on a heat map (Fig 2C–2D). Many of the compounds showed large numbers of up-regulated genes (Fig 2C-Decitabine, Givinostat, ML228, Alexidine and A485). Decitabine and givinostat are expected to reduce DNA methylation and increase histone acetylation respectively and thus would be expected to induce gene up-regulation. ML228 is a hypoxia inducible factor (HIF) activator and thus would be expected to up-regulate HIF target genes [27]. A485 has p300 inhibitor activity, which would lead to decreased histone acetylation – thus it was a surprise to see high numbers of up-regulated genes. CPI203 (BET inhibitor) showed the greatest number of down-regulated genes (Fig 2D). BET inhibitors are hypothesized to inhibit the assembly of transcription complexes mediated by BET domain-containing proteins binding to

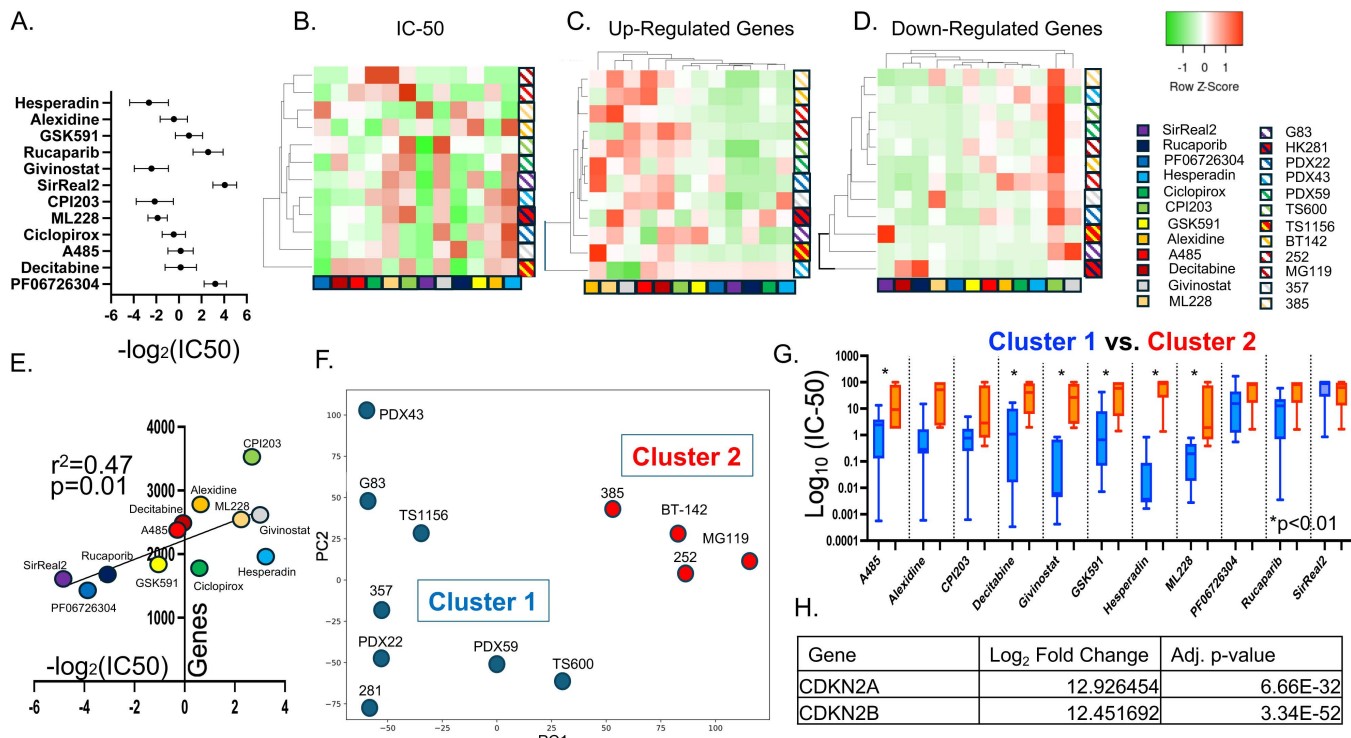

**Fig 2. A-B. The twelve previously identified compounds were tested against a new panel of twelve glioblastoma primary cultures.** Cultures were exposed for seven days and viability was tested at four different dosages (10uM, 1uM, 100nM and 10nM) with three replicates and an IC-50 was calculated using logistic regression. The mean and standard deviation of the -log$_2$(IC-50) is plotted. A heatmap of the IC-50's was constructed normalizing for each compound. C-D. Cultures were exposed to the twelve compounds for 7 days before undergoing RNA-seq analysis. The number of up-regulated (C.) and down-regulated (D.) genes compared to control were tabulated and shown as a heatmap. C. Alexidine, ML228, Givinostat, A485 and Decitabine showed a high number of up-regulated genes. D. CPI203 showed a high number of down-regulated genes. E. The number of up-regulated/down-regulated genes and the IC-50s for each drug were averaged across all lines and plotted as a scatterplot showing a correlation between larger genetic effects and lower IC-50s. F. The twelve cell lines underwent PCA analysis based on untreated RNA-seq data and revealed two clusters (Cluster 1 and Cluster 2). G. The average IC-50s were calculated for Cluster 1 and 2 are shown above showing higher drug resistance (e.g., higher IC-50) in Cluster 2 across multiple compounds. H. Differential expression analysis between cluster 1 and 2 found CDKN2A and CDKN2B to be the most differentially expressed genes (expression is higher in cluster 1).

acetylated lysine residues. The average IC-50 and number of target genes (both up- and down-regulated) was calculated for each compound across the panel of 12 cell lines. There was a significant correlation between the number of target genes and drug sensitivity (i.e., lower IC-50) indicating that compounds with larger effects on the transcriptome were also more potent at slowing cell growth (Fig 2E).

The observation that the IC-50s were so variable across cell lines raises the practical clinical concern of drug selection in an unknown tumor. With that in mind, we pursued the question of whether a cell line's baseline transcriptome could predict drug response. PCA clustering analysis of the baseline RNA sequencing data of each of the twelve lines revealed two clusters (Fig 2F). Comparing the average IC-50's for those two clusters revealed that for six of the twelve drugs, cluster 2 was significantly more resistant than cluster 1 (Fig 2G). Differential expression analysis between the two groups identified CDKN2A and CDKN2B as the most differentially expressed genes (higher in Cluster 1-drug sensitive group)(Fig 2H). CDKN2A [28] (which includes the p16/IKN4A and p14/ARF proteins) and CDKN2B [29] (INK4B) are well-known tumor suppressors. These types of experiments suggest that a clinically useful classification scheme of brain tumors could help drive drug selection and treatment decisions.

## CPI203 down-regulates a consistent set of DNA synthesis genes

Informed by the finding that candidate compounds could be classified as gene up-regulating or down-regulating, we next explored the question of whether the targets of these compounds were consistent across cell lines using intersection/overlap analysis. Lists of genes showing ≥2-fold change in response to drug treatment were generated and compared for overlapping genes across the glioblastoma spheres. Representative examples of resulting intersection gene sets are shown (Fig 3A–3C). When examining the target genes of each compound across the various cell lines, the overlapping sets were relatively small. Surprisingly, the largest gene sets were usually specific to one unique cell line/compound combination implying that each compound had different gene targets in each cell line(Green boxes Fig 3A–3C). In contrast, when examining the intersection of gene sets of different drugs within a single cell line, most cell lines showed a set of intersecting genes that were non-selectively up regulated in response to every drug (Fig 3B, blue box). It is possible that this gene set contains non-specific "toxicity genes." Interestingly, while many of the lines showed this phenomenon of a gene set non-specifically up-regulating in response to every drug, it was a different gene set for each cell line.

CPI203 was the only compound that induced a consistent set of target genes across the vast majority of cell lines (Fig 3C red box). Gene ontology analysis revealed that the consistent set of down-regulated genes were highly enriched for DNA synthesis genes (Fig 3D).

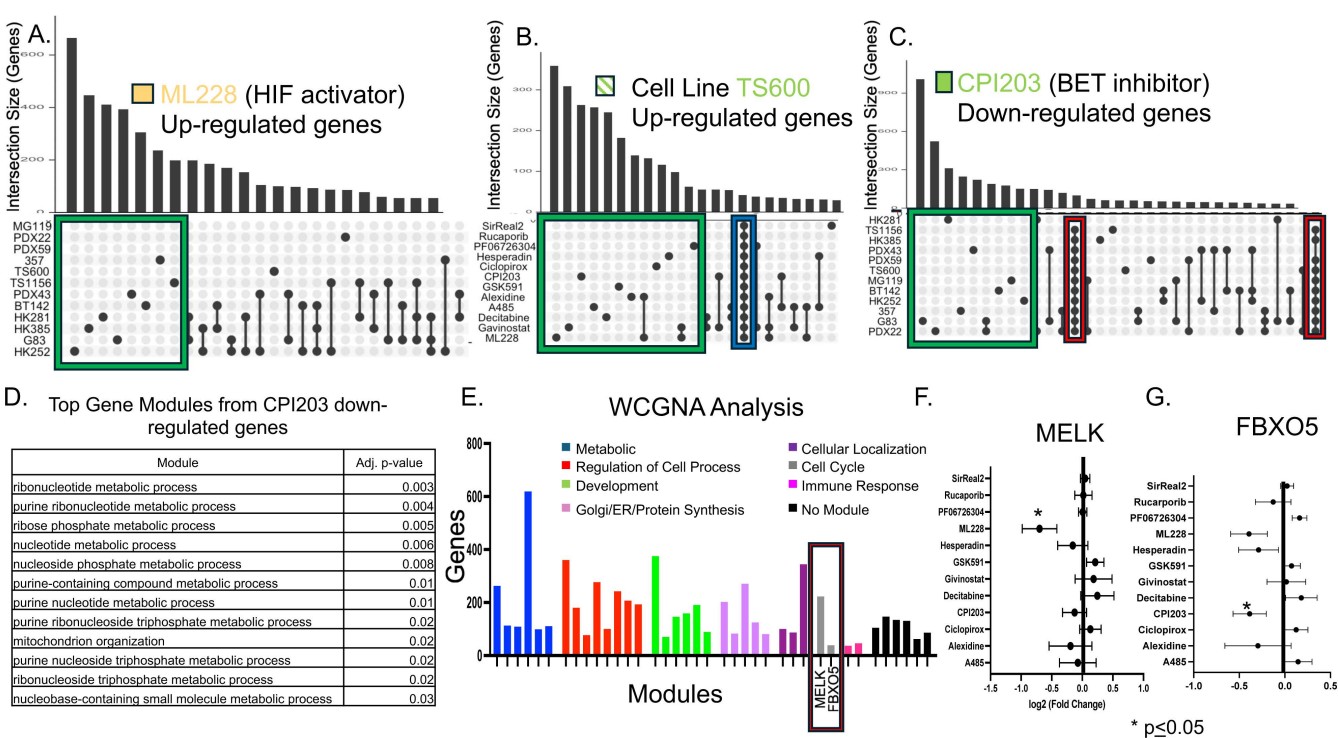

**Fig 3. A-C. Up-regulated and down-regulated genes (≥2-fold) were identified in each cell line/compound combination when compared to control (no drug).** Overlap analysis was performed to identify gene sets that were consistent in one drug across several cell lines or consistent in one cell line across several drugs. Most up-regulated and down-regulated genes were unique to one specific cell line/compound combination (green boxes) with two exceptions. First, in several of the cell lines there was a set of universally up-regulated genes (blue box) that are suspected to be "stress response" genes. Second there was a set of 103 genes that were consistently down-regulated by CPI203 (red box). D. Gene ontology was performed on those 103 down-regulated CPI203 genes and they were found to be highly enriched for DNA synthesis modules. D. WCGNA analysis was performed on 156 RNA-seq datasets to identify co-varying gene modules. 35 gene modules were identified and classified by gene ontology. Two of the modules were enriched for cell cycle genes (gray bars/red box) and were found to include the MELK and FBXO5 genes. E-F. RNA-seq data was re-analyzed to determine which compound had the greatest down-regulation effect on MELK and FBXO5 across all the cell lines.

## WCGNA analysis reveals MELK and FBXO5 as highly targetable gene modules in glioblastoma

Genes rarely act in isolation but rather function as part of a network. To determine targetable and variable gene networks we employed weight correlation gene network analysis (WCGNA) to identify sets of co-varying genes. Genes with a high level of covariance across different cell lines and that correlate under different drug conditions are assumed to be involved in the same pathway or network. This WCGNA analysis identified 39 gene modules in our dataset (Fig 3E). These gene modules then underwent gene ontology analysis and based on the top three gene ontology terms were divided into seven different cellular processes: metabolic, regulation of cell process, development, Golgi/ER protein synthesis, cellular localization, cell cycle and immune response. Of note, six of the modules had no significant gene ontology terms (black bars). Two of these gene modules were assigned to the cell cycle group (grey bars) and were heavily enriched for mitotic and cell cycle genes. One of these modules contained the "MELK" gene and the other contained the "FBXO5" gene. Both MELK and FBXO5 are heavily associated with cancer [30–33] in general and glioma in particular [34–39]. Using the previous RNA-seq data we analyzed which of the twelve compounds was most effective at reducing expression of these genes. ML228 was the most effective at reducing MELK expression and CPI203 was the most effective at reducing FBXO5 expression (Fig 3F–3G). Comparing these two modules (MELK and FBXO5) with publicly available gene expression data sets (lincsproject.org) revealed significant overlap between our MELK module and genes found in MELK over-expression experiments (Supplementary Figure 3). This implies that this module is at least partially regulated by MELK expression. There are several pharmacological inhibitors of MELK [35] and thus more efficient ways to target this module. In contrast, while LINCS expression data did confirm that genes in the FBXO5 module were correlated in other cell types, we were not able to identify a master regulator of the module (Supplementary Figure 3). This highlights the importance of the ability of chromatin-modifying compounds to regulate important gene pathways that might not otherwise be targetable. Given this finding we elected to focus on the FBXO5 module and created a correlation network figure of the twenty genes most strongly correlated with FBXO5. The distance of each spoke represents correlation across the entire dataset including all drugs and all lines (Fig 4A). The vast majority of these genes were related to cell division and DNA replication. In order to look at how each candidate compound affected the FBXO5 module, the effect of each compound on each gene in the module was averaged and displayed in a heatmap (Fig 4B). While CPI203 was the most effective compound at reducing FBXO5 gene expression specifically, MLL5 had a larger effect on the FBXO5 gene module as a whole. Testing the robustness of this finding on other BET inhibitors, two additional BET inhibitors (JQ1 and OTX015) were tested on four glioblastoma cell lines (Fig 4C) and showed consistent down-regulation of the FBXO5 module. BET inhibitors are hypothesized to disrupt the recruitment of transcription machinery and thus decrease gene expression. ATAC-seq is a sequencing technique that identifies open and accessible regions of chromatin (e.g., "peaks"). Using ATAC-seq, we tested the effect of three BET inhibitors (CPI203, JQ1, and OTX015) on a panel of four glioblastoma lines (357, 372, 385 and 412) to determine the effect on the number of peaks. Consistent with predictions, exposure to any of the three BET inhibitors led to a significant loss of peaks (i.e., closing of chromatin) globally (Fig 4D). Focusing more specifically at the FBXO5 module, we identified several peaks associated with the 20 most highly correlated genes. Again, BET inhibitor treatment led to significant decreases in many of those peaks implying more closed chromatin(Fig 4E).

## Discussion

Glioblastoma is a highly infiltrative disease that can use white matter tracts to travel to all areas of the brain, including the opposite hemisphere. As such, local therapies such as surgery and radiation are insufficient to establish long-term disease control [1]. Chemotherapy drugs are divided into three categories. First, cytotoxic compounds (e.g., temozolomide, paclitaxel, carmustine) largely inhibit processes of cellular divisions and thus disproportionately affect dividing cells. Second, molecularly targeted inhibitors are designed to inhibit specific pro-growth molecules, frequently tyrosine kinase enzymes, e.g., imatinib [40,41], gefitinib [42] and vorasidenib. Lastly and most recently, monoclonal antibodies that interfere with tumor vascularization or immune-suppressing checkpoint mediators [43] have the potential to

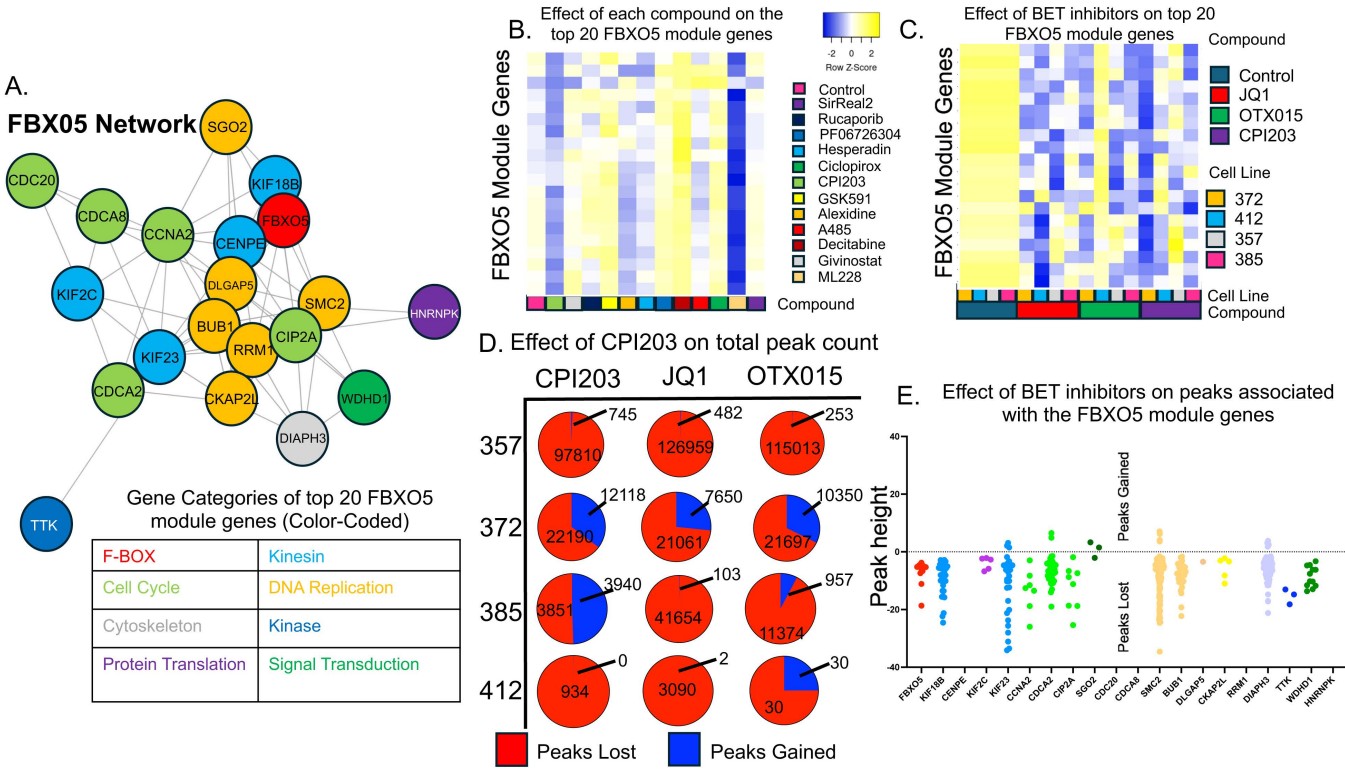

**Fig 4. A. A "hub-spoke" plot was created using the 20 genes most correlated with FBXO5 expression.** Shorter spoke distance implies stronger correlation. Genes were categorized and color-coded by gene ontology. B. The average fold-change for each of the 20 highly correlated genes was calculated for each drug across all twelve cell lines and plotted as a heatmap showing the most down-regulation in ML228 and CPI203. C. To confirm these findings the experiment was repeated with two more BET inhibitors (OTX015 and JQ1) and an additional four cell lines (272, 412, 357, 385). D. These three drugs (JQ1, OTX015 and CPI203) and four lines (357, 372, 385 and 412) were additionally subjected to ATAC-seq analysis. The number of total peaks lost and gained globally was calculated. E. ATAC-seq peaks associated with the 20 most highly correlated FBXO5 genes were identified. Those peaks that showed a significant height change in response to drug treatment were combined and plotted.

reduce nutrient delivery [14,15,44] and/or increase the host's immune response against tumor cells. To establish a new modality for glioblastoma therapy centered on reprogramming the epigenome, we evaluated 106 drug candidates using three-dimensional gliomasphere lines. Recent evidence suggests that cytotoxic drug sensitivity using low passage patient-derived glioblastoma cells correlates with in vivo clinical drug sensitivity [25] and thus in vitro models can be a useful tool to both test potential new compounds and guide patient-specific drug treatment. Thus, in this study the authors use a relatively large panel of patient-derived cell lines to identify a promising "gene modulator" compound for further testing and a potential clinical trial.

While glioblastoma is a highly heterogenous disease, standard of care radiation and chemotherapy protocols for glioblastoma are relatively limited and rigid with treatment frequently limited to a single chemotherapy agent (temozolomide). The efficacy of temozolomide varies across different glioblastomas with known predictors of response (e.g., MGMT methylation status). Surprisingly, temozolomide is still the standard treatment even when molecular markers predict a poor response (i.e., unmethylated MGMT promoter). Consistent with the variable response to temozolomide, this study also found a significant amount of heterogeneity in the cytotoxic response to agents targeting the epigenome across the different glioblastomas. Further, there was similar heterogeneity in gene expression responses to the agents across the different glioblastomas. For most of the tested drugs, the gene targets in a given cell line were largely unique and distinct

from the gene targets in the other lines. CPI203 was an exception to this trend with a large set of common target genes inhibiting DNA replication.

One approach to addressing the problem of variable drug efficacy is to identify predictors of response. In the case of gliomas, some predictors have been identified. MGMT methylation predicts greater efficacy of temozolomide [45] and 1p/19q deletion predicts a beneficial response to radiation [46]. Other tumor types have used expression profiles (e.g., medulloblastoma) or histology findings (craniopharyngioma, lymphomas) to subdivide tumors into subtypes that are associated with prognosis and drug response [47,48]. This approach has had mixed results in glioma. Initial attempts using expression analysis yielded four groups [2,4]. However, when controlling for the IDH mutation, each group had similar prognoses and treatment response profiles [2,49]. Further, single-cell RNA-seq analysis revealed that instead of being a collection of cells from one subtype, tumors contained a mixture of cells from each subtype [50]. Given the potential value of tumor categorization to guide treatment decisions, it seems logical to subdivide tumors based on drug sensitivity and treatment response. Although this study was not sufficiently powered to identify specific sensitivity gene expression profiles for each compound, clustering on baseline gene expression did identify two clusters that differed in drug response and revealed a genetic biomarker (CDKN2A and CDKN2B) (Fig 3). A larger study with more cell lines would potentially be able to identify gene signatures for individual drugs. Further, it may be possible to create a more useful tumor classification based on drug response profiles.

A significant amount of glioma research has focused on the deletion of critical tumor suppressor genes, e.g., p53, PTEN, NF1, and Rb. While these deletion events are essential elements of tumorigenesis, they are not amenable to pharmacological intervention. Less attention has focused on malignant gene programs of gliomas. In this study, we used weighted gene co-expression network analysis (WGCNA) to find gene networks that could be targeted with the compounds tested. This analysis identified two "cell cycle" modules one including MELK and the other FBXO5 (also known as "early mitotic inhibitor 1" EMI1). MELK is a serine/threonine kinase with a reported role in driving growth in glioblastoma [34–36]. This MELK module appears to largely be regulated by MELK expression. Fortunately, a selective MELK inhibitor has been developed (OTS167) and is already in phase I clinical trials(Clinical Trial ID: NCT02926690). Regulation of the FBXO5 module was more complicated and we were unable to identify a master regulator of this module. This complexity highlights the importance of chromatin-modifying compounds to down-regulate these more complex gene programs where a single gene inhibitor does not exist.

While this study presents helpful information for the development of epigenome-modulating compounds for glioma therapy, there are several weaknesses that should be discussed. While the list of gene modulating compounds was extensive it was not exhaustive and other promising compounds may have been excluded. Second, due to the variability in potency among different compounds and different cell lines, the drug-treated gene expression data by necessity was acquired at non-uniform dosages. Third, PCA analysis of the gene expression data divided the cohort of cell lines into two clusters with different drug sensitivity profiles. Three of the four cell lines in cluster 2 contain the IDH1 mutation and this may have confounded the results.

In conclusion, gene modulating compounds are a promising addition to the complement of cyto-toxic and immune therapy compounds currently in clinical use. Of the compounds tested, CPI203 (BET inhibitor) had the most consistent set of target genes and was effective at down-regulating DNA replication genes, and down-regulating the expression of a potentially important oncogenic FBXO5 gene module.

## Methods

### Initial in vitro drug screen

A drug screen of 106 known epigenetic compounds targeting 36 genes was conducted against five glioblastoma lines mutant (357, 372, 385, 408, and 412) for three days (1uM, 10uM) and twenty-eight days (500nM) with two replicates. The cells were a gift from Dr. Harley Kornblum and are characterized in Supplementary Figure 1 [51]. Initial cell numbers were

plated to maximize the number of viable cells at day the target time point without over-growth. Additional drug and growth factor were added on a weekly basis. At the end of each experiment, the number of viable cells was estimated by MTT at each dose and was used to create an "Area under the curve" value (AUC). To identify compounds that worked to lower viability over time and rule out compounds with non-specific cyto-toxicity we calculated (delta AUC = Viability AUC day 3 – Viability AUC day 28). Drugs were then grouped according to their corresponding target genes (e.g., BET, HDAC, DNMT) and ranked by delta AUC (Higher Delta AUC implies more anti-tumor efficacy). Within each group, the ranks of all the compounds were averaged to create an "average rank" for the group. The top twelve compound groups were identified. Within each compound group the compound with the highest delta AUC was identified for further testing.

## 12 cell lines x 12 drugs

After the initial drug screen, the twelve most promising compounds were selected from the twelve most promising drug classes. Cells were plated at numbers to maximize MTT signal at 7 days and treated with four drug doses (10nM, 100nM, 1uM, 10uM) with three replicates. MTT was obtained on day 7 and IC-50 was calculated based on logistic regression Graphpad version 10.1.1 for Mac OS (www.graphpad.com). Each cell line was then treated with each of the twelve compounds at a dose chosen to obtain 65% viability. In most conditions cells were treated at 500nM however in a few cases it was required to reduce the dose to 250nM to obtain enough viable material (supplementary figure 2). The cells were then harvested on day 7 and prepared for RNA-seq analysis. There was one RNA-seq data set per condition (156 RNA-seq data sets total).

**RNA sample preparation and RNA-seq data analysis.** RNA (≥200 nucleotides) was purified from cells using the Quick-RNA MiniPrep Plus Kit (Zymo Research, #R1057). The quality control check on RNA-seq reads was performed with FastQC v0.11.7. Adapter sequences and bad quality segments were trimmed using Trim Galore! v0.4.2 and cutadapt v1.9.1. The trimmed reads were aligned to the reference human genome version GRCh38/hg38 with the program STAR v2.6.1e. Duplicate aligned reads were removed using the program sambamba v0.6.8. Gene-level expression was assessed by counting features for each gene, as defined in the NCBI's RefSeq database. Read counting was done with the program featureCounts v1.6.2 from the Rsubread package. Raw counts were normalized with respect to library size and transformed to log2 scale using rlog() function in R package DESeq2 v1.26.0.

**ATAC-Seq**: ATAC-seq libraries were created as described previously [52]. Briefly, 50,000 cells were spun down at 500 x g for 5 minutes at 4°C. Cells were resuspended in ATAC-resuspension buffer containing 0.1% NP40, 0.1% tween-20, and 0.01% digitonin and incubated on ice for 3 minutes. Cells were washed with ATAC-resuspension buffer containing 0.1% tween-20. Nuclei were pelleted at 500 x g at 4C for 10 minutes. Nuclei were resuspended in transposition reaction mix containing TD 2x reaction buffer (Illumina, #20034197), TDE1 Nextera Tn5 Transposase (Illumina, #20034197), and nuclease free water. The reaction was incubated at 37°C in a water bath for 30 minutes. Immediately following transposition, DNA was purified with a Qiagen MinElute PCR Purification Kit (Qiagen, #28004). Following purification, transposed DNA was mixed with NEBNext high-Fidelity 2x PCR Master Mix (NEB, #M0541S), AD1_noMX and AD2.1–2.16 barcoded primers, and nuclease free water. Samples were amplified for 12 cycles. Immediately following amplification, DNA was purified using a Qiagen MinElute PCR Purification Kit. ATAC-seq reads in FASTQ format were first subjected to quality control to assess the need for trimming of adapter sequences or bad quality segments. The programs used in these steps were FastQC v0.11.7 and Trim Galore! v0.6.6. The trimmed reads were aligned to the reference human genome version GRCh38/hg38 via bowtie2 v2.5.1 with the '--very-sensitive' setting. Aligned reads were stripped of mitochondrial and duplicate reads using the samtools v.1.17 toolkit. Peaks were called using MACS2 v2.2.9.

**Weight correlation gene network analysis (WCGNA).** In this study, Weighted Gene Co-expression Network Analysis (WGCNA) was employed to identify co-expressed gene modules from RNA sequencing data. We used the normalized RNA-seq data to construct a similarity matrix based on the Pearson correlation of gene expression profiles across samples. To emphasize strong correlations and minimize the impact of noise, this matrix was transformed into a weighted

network using a power function. The choice of the soft-thresholding power was based on the criterion of approximate scale-free topology. After determining the appropriate power value, we converted the similarity matrix into a topological overlap matrix (TOM), which was subsequently used to cluster genes into modules via average linkage hierarchical clustering. The minimum module size was set to 30 to define the modules. The entire analysis was conducted using the "WGCNA" 1.72 package in R.

## Supporting information

**Supplementary Figure 1: A. Five glioblastoma lines underwent a drug screen of 106 compounds with viability measured on day 3 (1uM and 10uM) and day 28 (500nM).** Delta AUC = AUC Day 3-AUC Day 28. The average delta AUC values for each compound were ranked and are shown. B. This study uses a panel of 14 glioblastoma/glioma lines that have been partially characterized.
(PDF)

**Supplementary Figure 2: Twelve cell lines were exposed to one of the twelve compounds for seven days before undergoing RNA-seq analysis.** Initial dose was 500nM however it was necessary to reduce the dose in some cases due to excessive cytotoxicity.
(PDF)

**Supplementary Figure 3: A. The 223 genes identified in the MELK module of the WCGNA analysis are listed.** B. The 39 genes identified in the FBXO5 module of the WCGNA analysis are listed. C. Overlap analysis shows a highly significant overlap between the MELK module and genes identified in MELK over-expression experiments (lincsproject.org). Overlap analysis between FBXO5 module and genes identified in FBXO5 over-expression or knock-down experiments shows no overlap. There was significant overlap between FBXO5 module genes and genes that are closely correlated with FBXO5 in publicly available expression data.
(PDF)

## Acknowledgments

We thank the Genomics, Epigenomics and Sequencing Core at the University of Cincinnati for their assistance and guidance in experimental design and data optimization.

We thank the Biostatistics and Bioinformatics Core at the University of Colorado Anschutz Medical Campus for their expertise and guidance on bio-informatic analysis and data presentation.

We thank the Kornblum Lab at UCLA for their generous donation of cell lines with associated expression and genetic profiles.

## Author contributions

**Conceptualization:** Matthew Garrett, Troy Carnwath, Mark Hoeprich, Marilyn Reed, David R. Plas.

**Data curation:** Matthew Garrett, Troy Carnwath, Rebecca Albano, Catherine A. Behrmann, Merissa Pemberton, Farah Barakat, Eric O'Brien, Brett VanCauwenbergh.

**Formal analysis:** Matthew Garrett, Troy Carnwath, Yonghua Zhuang, Mario Medvedovic, David R. Plas.

**Funding acquisition:** Matthew Garrett, Troy Carnwath, Rebecca Albano, Robert Lober, David R. Plas.

**Investigation:** Matthew Garrett, Catherine A. Behrmann, Farah Barakat, Eric O'Brien, David R. Plas.

**Methodology:** Matthew Garrett, Troy Carnwath, Yonghua Zhuang, Catherine A. Behrmann, Eric O'Brien, Brett VanCauwenbergh, David R. Plas.

**Project administration:** Matthew Garrett, John Perentesis, David R. Plas.

**Resources:** Matthew Garrett, Robert Lober, Marilyn Reed, David R. Plas.

**Supervision:** Matthew Garrett, Robert Lober, Daniel Woo, John Perentesis, Mario Medvedovic, David R. Plas.

**Visualization:** Matthew Garrett.

**Writing – original draft:** Matthew Garrett.

**Writing – review & editing:** Matthew Garrett, Mark Hoeprich, Anthony Paravati, Hailey Spry, Rani Nasser, David R. Plas.

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
