## [Decision Letter · Decision Letter 0]

31 Jul 2024

PONE-D-24-24126CPI203, a BET inhibitor, down-regulates a consistent set of DNA synthesis genes across a wide array of glioblastoma lines.PLOS ONE

Dear Dr. Garrett,

Thank you for submitting your manuscript to PLOS ONE. After careful consideration, we feel that it has merit but does not fully meet PLOS ONE’s publication criteria as it currently stands. Therefore, we invite you to submit a revised version of the manuscript that addresses the points raised during the review process.

Detailed comments from reviewers are:

Reviewer#1:

The study demonstrates that CPI203, a BET inhibitor, effectively downregulates DNA synthesis genes across different glioblastoma lines. This finding offers a promising therapeutic approach for glioblastoma, potentially impacting the treatment of this aggressive disease. Additionally, the authors identified CDKN2A and CDKN2B as biomarkers for determining CPI203 sensitivity in glioblastoma lines through PCA clustering and differential expression analyses. Furthermore, using WCGNA, the authors discovered that the FBXO5 oncogenic module was downregulated by CPI203 treatment.

Q1. Validation of CDKN2A and CDKN2B as Biomarkers of CPI203 Sensitivity

The authors claim that high expression of CDKN2A and CDKN2B leads to higher CPI203 sensitivity. To substantiate this claim, it would be beneficial to perform genetic knockout or knockdown experiments using CRISPRko, CRISPRi, or RNAi targeting CDKN2A, CDKN2B, or both. These experiments should demonstrate that the loss of these genes results in reduced sensitivity to CPI203, thereby validating them as biomarkers.

Q2. Concerns about the General Toxicity of BET Inhibitor (CPI203)

The BET family of proteins includes four conserved mammalian members: BRD2, BRD3, BRD4, and BRDT. According to DepMap (24Q2), BRD4 is a common essential gene, raising general toxicity concerns about using CPI203, a BET inhibitor, in clinical applications. Can the authors address these concerns and discuss the potential toxicity of CPI203 in clinical settings?

Q3. Utilizing DepMap Data for Drug Sensitivity Profiles

On page 19 of the manuscript, the authors mention that a larger study with more cell lines could potentially identify gene signatures for individual drugs and create a more useful tumor classification based on drug response profiles. Public data from DepMap on the drug sensitivity profile of CPI203 across 472 cancer cell lines is available (source: https://depmap.org/portal/compound/CPI-203?tab=overview). The authors should consider utilizing this data to enhance their study and develop a more comprehensive tumor classification based on drug response profiles.

Q4. Data Availability and Code Reproducibility

The authors agreed to make all data fully available without restriction in the submission package. However, their GitHub repository currently provides links to the Illumina BaseSpace account, which requires login to retrieve the data. To ensure full public accessibility, the authors should upload their data to the Gene Expression Omnibus (GEO). Additionally, uploading their R/Python codes to the GitHub repository would make their major analysis workflows transparent and reproducible.

Reviewer#2

Summary: Glioblastoma is a complex disease that exhibits heterogeneity between patients and between tumors. To date, the standard of care treatment for glioblastoma is temozolomide. The varying response of temozolomide shows the need to develop new therapeutic targets and their associated compounds for glioblastoma treatment. Glioblastoma is propagated by the expression of pro-tumor growth factors (ie. EGFR) or deletion of tumor suppression genes (ie. PTEN, p53, Rb), the authors’ novelty utilizes chromatin modifying agents to identify new treatment options in glioblastoma. To identify viable chromatin modifying/epigenetic compounds as treatment options for glioblastoma, the authors screened 106 chromatin modifying/epigenetic compounds targeting 36 different genes first in independently derived gliomaspheres, lead compounds for each of the 12 compound class underwent a secondary screening using early passage of additional twelve gliomasphere lines. The authors calculated “delta AUC” to identify lead compound for each compound class. Of all compounds tested, CPI203, a BET class inhibitor reduced cell viability the most. RNA-seq of compound treated vs nontreated cells revealed that the same compound showed varying response in different cell line. However, CPI203, a BET class inhibitor shows the most consisted gene disruption in different cell lines. Further, the authors performed weighted correlation gene net analysis (WCGNA) ot identify sets of co-varying genes after compound treatment because genes often function through a network. WCGNA identified enriched genes “MELK” and “FBXO5” in sensitive cells, both of which as involved in mitotic and cell cycle, suggesting these two genes may serve as new therapeutic targets for glioblastoma treatment. Lastly, the author performed ATAC-seq in glioblastoma cell line after treatment with lead BET inhibitor(CPI203, JQ1, and OTX015). ATAC-seq showed a reduction of gene accessibility in cell lines after BET inhibitor treatment suggesting specificity of lead compounds.

Comments

1. In panel 1. The authors performed 106 compound screening in 5 cell lines. Each cell line was treated with 1uM or 10 uM for three days and 500 nM for twenty-eight days. Lead compound was identified by calculating “delta AUC” by the difference of day three and day twenty-eight by MTT assay.

a. In the twenty-eight day treatment, were the cells passaged as they become confluent? If cells were passaged, then the conditions of the twenty-eight day treatment would be different compared to that of three day treatment. If the goal was to identify lead compound by three days in each cell lines, why not use a vehicle control such as DMSO or solvent used to dilute each compound, then compare cell viability at day three to the vehicle control.

Minor Comments

1. In figure 2F, the authors performed RNA-seq in compound treated vs control of each cell lines. PCA clustering analysis revealed two cluster of sensitive and resistant cells. Cluster 1, the sensitive cluster seems to have varying transcriptome across PCA1 whereas the cells in cluster 2 are more similar. Gene expression of cluster 2 may reveal additional gene sets in cluster 2 that can explain drug resistance.

a. Cluster 1 was observed to express higher well-known tumor suppressor genes (CHKN2A/CDKN2B), could CPI203 serve as conditional therapy?

b. Cluster 1 seems to have varied expression across PCA1. Could this suggest that CPI203 is binding to a specific target rather than a transcriptome similarity?

2. The authors made mentioned that the compounds used in the study were extensive but not exhaustive. Perhaps the authors should include rationale of why these compounds or specifically 106 was chosen.

We look forward to receiving your revised manuscript.

Kind regards,

Mingli Li

Academic Editor

PLOS ONE

Journal Requirements:

2. PLOS requires an ORCID iD for the corresponding author in Editorial Manager on papers submitted after December 6th, 2016. Please ensure that you have an ORCID iD and that it is validated in Editorial Manager. To do this, go to ‘Update my Information’ (in the upper left-hand corner of the main menu), and click on the Fetch/Validate link next to the ORCID field. This will take you to the ORCID site and allow you to create a new iD or authenticate a pre-existing iD in Editorial Manager. Please see the following video for instructions on linking an ORCID iD to your Editorial Manager account: https://www.youtube.com/watch?v=_xcclfuvtxQ".

4. We notice that your supplementary figures are included in the manuscript file. Please remove them and upload them with the file type 'Supporting Information'. Please ensure that each Supporting Information file has a legend listed in the manuscript after the references list.

5. Please remove your figures from within your manuscript file, leaving only the individual TIFF/EPS image files, uploaded separately. These will be automatically included in the reviewers’ PDF.

Additional Editor Comments:

This manuscript is well-written. The experiments were well-designed. Taken together with two reviewers' comments, we would like to request minor revision before publishing. Detailed comments from reviewers are:

Reviewer#1:

The study demonstrates that CPI203, a BET inhibitor, effectively downregulates DNA synthesis genes across different glioblastoma lines. This finding offers a promising therapeutic approach for glioblastoma, potentially impacting the treatment of this aggressive disease. Additionally, the authors identified CDKN2A and CDKN2B as biomarkers for determining CPI203 sensitivity in glioblastoma lines through PCA clustering and differential expression analyses. Furthermore, using WCGNA, the authors discovered that the FBXO5 oncogenic module was downregulated by CPI203 treatment.

Q1. Validation of CDKN2A and CDKN2B as Biomarkers of CPI203 Sensitivity

The authors claim that high expression of CDKN2A and CDKN2B leads to higher CPI203 sensitivity. To substantiate this claim, it would be beneficial to perform genetic knockout or knockdown experiments using CRISPRko, CRISPRi, or RNAi targeting CDKN2A, CDKN2B, or both. These experiments should demonstrate that the loss of these genes results in reduced sensitivity to CPI203, thereby validating them as biomarkers.

Q2. Concerns about the General Toxicity of BET Inhibitor (CPI203)

The BET family of proteins includes four conserved mammalian members: BRD2, BRD3, BRD4, and BRDT. According to DepMap (24Q2), BRD4 is a common essential gene, raising general toxicity concerns about using CPI203, a BET inhibitor, in clinical applications. Can the authors address these concerns and discuss the potential toxicity of CPI203 in clinical settings?

Q3. Utilizing DepMap Data for Drug Sensitivity Profiles

On page 19 of the manuscript, the authors mention that a larger study with more cell lines could potentially identify gene signatures for individual drugs and create a more useful tumor classification based on drug response profiles. Public data from DepMap on the drug sensitivity profile of CPI203 across 472 cancer cell lines is available (source: https://depmap.org/portal/compound/CPI-203?tab=overview). The authors should consider utilizing this data to enhance their study and develop a more comprehensive tumor classification based on drug response profiles.

Q4. Data Availability and Code Reproducibility

The authors agreed to make all data fully available without restriction in the submission package. However, their GitHub repository currently provides links to the Illumina BaseSpace account, which requires login to retrieve the data. To ensure full public accessibility, the authors should upload their data to the Gene Expression Omnibus (GEO). Additionally, uploading their R/Python codes to the GitHub repository would make their major analysis workflows transparent and reproducible.

Reviewer#2

Summary: Glioblastoma is a complex disease that exhibits heterogeneity between patients and between tumors. To date, the standard of care treatment for glioblastoma is temozolomide. The varying response of temozolomide shows the need to develop new therapeutic targets and their associated compounds for glioblastoma treatment. Glioblastoma is propagated by the expression of pro-tumor growth factors (ie. EGFR) or deletion of tumor suppression genes (ie. PTEN, p53, Rb), the authors’ novelty utilizes chromatin modifying agents to identify new treatment options in glioblastoma. To identify viable chromatin modifying/epigenetic compounds as treatment options for glioblastoma, the authors screened 106 chromatin modifying/epigenetic compounds targeting 36 different genes first in independently derived gliomaspheres, lead compounds for each of the 12 compound class underwent a secondary screening using early passage of additional twelve gliomasphere lines. The authors calculated “delta AUC” to identify lead compound for each compound class. Of all compounds tested, CPI203, a BET class inhibitor reduced cell viability the most. RNA-seq of compound treated vs nontreated cells revealed that the same compound showed varying response in different cell line. However, CPI203, a BET class inhibitor shows the most consisted gene disruption in different cell lines. Further, the authors performed weighted correlation gene net analysis (WCGNA) ot identify sets of co-varying genes after compound treatment because genes often function through a network. WCGNA identified enriched genes “MELK” and “FBXO5” in sensitive cells, both of which as involved in mitotic and cell cycle, suggesting these two genes may serve as new therapeutic targets for glioblastoma treatment. Lastly, the author performed ATAC-seq in glioblastoma cell line after treatment with lead BET inhibitor(CPI203, JQ1, and OTX015). ATAC-seq showed a reduction of gene accessibility in cell lines after BET inhibitor treatment suggesting specificity of lead compounds.

Comments

1. In panel 1. The authors performed 106 compound screening in 5 cell lines. Each cell line was treated with 1uM or 10 uM for three days and 500 nM for twenty-eight days. Lead compound was identified by calculating “delta AUC” by the difference of day three and day twenty-eight by MTT assay.

a. In the twenty-eight day treatment, were the cells passaged as they become confluent? If cells were passaged, then the conditions of the twenty-eight day treatment would be different compared to that of three day treatment. If the goal was to identify lead compound by three days in each cell lines, why not use a vehicle control such as DMSO or solvent used to dilute each compound, then compare cell viability at day three to the vehicle control.

Minor Comments

1. In figure 2F, the authors performed RNA-seq in compound treated vs control of each cell lines. PCA clustering analysis revealed two cluster of sensitive and resistant cells. Cluster 1, the sensitive cluster seems to have varying transcriptome across PCA1 whereas the cells in cluster 2 are more similar. Gene expression of cluster 2 may reveal additional gene sets in cluster 2 that can explain drug resistance.

a. Cluster 1 was observed to express higher well-known tumor suppressor genes (CHKN2A/CDKN2B), could CPI203 serve as conditional therapy?

b. Cluster 1 seems to have varied expression across PCA1. Could this suggest that CPI203 is binding to a specific target rather than a transcriptome similarity?

2. The authors made mentioned that the compounds used in the study were extensive but not exhaustive. Perhaps the authors should include rationale of why these compounds or specifically 106 was chosen.

Reviewers' comments:

Reviewer's Responses to Questions

**Comments to the Author**

1. Is the manuscript technically sound, and do the data support the conclusions?

Reviewer #1: Partly

Reviewer #2: Yes

2. Has the statistical analysis been performed appropriately and rigorously? 

Reviewer #1: Yes

Reviewer #2: Yes

3. Have the authors made all data underlying the findings in their manuscript fully available?

Reviewer #1: No

Reviewer #2: Yes

4. Is the manuscript presented in an intelligible fashion and written in standard English?

Reviewer #1: Yes

Reviewer #2: Yes

5. Review Comments to the Author

Reviewer #1: Review Comments to the Authors

The study demonstrates that CPI203, a BET inhibitor, effectively downregulates DNA synthesis genes across different glioblastoma lines. This finding offers a promising therapeutic approach for glioblastoma, potentially impacting the treatment of this aggressive disease. Additionally, the authors identified CDKN2A and CDKN2B as biomarkers for determining CPI203 sensitivity in glioblastoma lines through PCA clustering and differential expression analyses. Furthermore, using WCGNA, the authors discovered that the FBXO5 oncogenic module was downregulated by CPI203 treatment.

Q1. Validation of CDKN2A and CDKN2B as Biomarkers of CPI203 Sensitivity

The authors claim that high expression of CDKN2A and CDKN2B leads to higher CPI203 sensitivity. To substantiate this claim, it would be beneficial to perform genetic knockout or knockdown experiments using CRISPRko, CRISPRi, or RNAi targeting CDKN2A, CDKN2B, or both. These experiments should demonstrate that the loss of these genes results in reduced sensitivity to CPI203, thereby validating them as biomarkers.

Q2. Concerns about the General Toxicity of BET Inhibitor (CPI203)

The BET family of proteins includes four conserved mammalian members: BRD2, BRD3, BRD4, and BRDT. According to DepMap (24Q2), BRD4 is a common essential gene, raising general toxicity concerns about using CPI203, a BET inhibitor, in clinical applications. Can the authors address these concerns and discuss the potential toxicity of CPI203 in clinical settings?

Q3. Utilizing DepMap Data for Drug Sensitivity Profiles

On page 19 of the manuscript, the authors mention that a larger study with more cell lines could potentially identify gene signatures for individual drugs and create a more useful tumor classification based on drug response profiles. Public data from DepMap on the drug sensitivity profile of CPI203 across 472 cancer cell lines is available (source: https://depmap.org/portal/compound/CPI-203?tab=overview). The authors should consider utilizing this data to enhance their study and develop a more comprehensive tumor classification based on drug response profiles.

Q4. Data Availability and Code Reproducibility

The authors agreed to make all data fully available without restriction in the submission package. However, their GitHub repository currently provides links to the Illumina BaseSpace account, which requires login to retrieve the data. To ensure full public accessibility, the authors should upload their data to the Gene Expression Omnibus (GEO). Additionally, uploading their R/Python codes to the GitHub repository would make their major analysis workflows transparent and reproducible.

Reviewer #2: The authors have provided evidence in support of the potential of chromatin modifying agents. In the study, the authors have shown drug efficacy and pathway of action. The identification of a specific drug target and binding specificity can greatly improve the study.

6. PLOS authors have the option to publish the peer review history of their article (what does this mean? ). If published, this will include your full peer review and any attached files.

**Do you want your identity to be public for this peer review?** For information about this choice, including consent withdrawal, please see our Privacy Policy .

Reviewer #1: No

Reviewer #2: **Yes: ** Benjamin Z Kuang

---

## [Author Response · Author response to Decision Letter 1]

13 Mar 2025

Thank you for your consideration of our manuscript “CPI203, a BET inhibitor, down-regulates a consistent set of DNA synthesis genes across a wide array of glioblastoma lines” for publication in the journal “PLOSone.” We thank the reviewers for taking their time to review our manuscript and submitting their revisions. We have incorporated these revisions into the next version of the manuscript and feel that it has made the manuscript much stronger. What follows is a point-by-point response to the reviewers’ critiques and how they have been incorporated into the current version.

This work is not currently under review anywhere else, and the authors have no conflicts of interest to disclose.

Sincerely,

Matthew Garrett M.D.,Ph.D.

Reviewer#1:

The study demonstrates that CPI203, a BET inhibitor, effectively downregulates DNA synthesis genes across different glioblastoma lines. This finding offers a promising therapeutic approach for glioblastoma, potentially impacting the treatment of this aggressive disease. Additionally, the authors identified CDKN2A and CDKN2B as biomarkers for determining CPI203 sensitivity in glioblastoma lines through PCA clustering and differential expression analyses. Furthermore, using WCGNA, the authors discovered that the FBXO5 oncogenic module was downregulated by CPI203 treatment.

Q1. Validation of CDKN2A and CDKN2B as Biomarkers of CPI203 Sensitivity

The authors claim that high expression of CDKN2A and CDKN2B leads to higher CPI203 sensitivity. To substantiate this claim, it would be beneficial to perform genetic knockout or knockdown experiments using CRISPRko, CRISPRi, or RNAi targeting CDKN2A, CDKN2B, or both. These experiments should demonstrate that the loss of these genes results in reduced sensitivity to CPI203, thereby validating them as biomarkers.

-The authors apologize for the lack of clarity regarding the CDKN2A/B genes and CPI203 sensitivity. The dataset shows that across all cell lines and drugs tested, there was a general clustering of sensitive and resistant cell lines. CDKN2A/B expression has associated with the more drug sensitive group. This makes sense as both genes are tumor suppressors and deletion of these genes has been shown to be associated with more aggressive behavior. The authors feel that we do not have statistical power to make claims about genetic biomarkers of drug sensitivity but that would be our next step, and we are actively collecting more glioma cell lines.

Q2. Concerns about the General Toxicity of BET Inhibitor (CPI203)

The BET family of proteins includes four conserved mammalian members: BRD2, BRD3, BRD4, and BRDT. According to DepMap (24Q2), BRD4 is a common essential gene, raising general toxicity concerns about using CPI203, a BET inhibitor, in clinical applications. Can the authors address these concerns and discuss the potential toxicity of CPI203 in clinical settings?

-This is an excellent point. After reviewing nine randomized clinical trials using reported “first-generation” BET inhibitors (e.g. OTX-015, CPI-061, GSK525762, and BAY1238097) , many of the patient reported gastrointestinal and hematological toxic side effects. It was suspected that this may have been due to short half-lives requiring frequent dosing. Second generation BET inhibitors (e.g. Trotaresib, ZEN-3694) are aiming to limit off-target toxicity. There is also some interest in synergistic therapy which may enable lower less toxic dosing (Karakashev et al. 2017). The text will be amended to add this point.

Q3. Utilizing DepMap Data for Drug Sensitivity Profiles

On page 19 of the manuscript, the authors mention that a larger study with more cell lines could potentially identify gene signatures for individual drugs and create a more useful tumor classification based on drug response profiles. Public data from DepMap on the drug sensitivity profile of CPI203 across 472 cancer cell lines is available (source: https://depmap.org/portal/compound/CPI-203?tab=overview). The authors should consider utilizing this data to enhance their study and develop a more comprehensive tumor classification based on drug response profiles.

-The Depmap portal from the Broad Insititute is an invaluable resource for drug sensitivity and genetic expression information with 472 lines including 34 brain/CNS lines. Unfortunately, in the case of CPI-203, there were only weak correlations between specific gene expression and drug sensitivity. In the case of CPI-203 the best gene was ZNF175 with a Pearson correlation of 0.24. Further, the expanded linear prediction model was not highly accurate. This is important information and highlights potential difficulties in identifying genetic signatures of drug response and has been included in the text.

Q4. Data Availability and Code Reproducibility

The authors agreed to make all data fully available without restriction in the submission package. However, their GitHub repository currently provides links to the Illumina BaseSpace account, which requires login to retrieve the data. To ensure full public accessibility, the authors should upload their data to the Gene Expression Omnibus (GEO). Additionally, uploading their R/Python codes to the GitHub repository would make their major analysis workflows transparent and reproducible.

-We absolutely agree with reviewer 1 and apologize for not making the data available. We have since uploaded the data to the Gene Expression Omnibus (GEO). The website is still in the process of validating the data and we’ll include a link when available. In the interim we have also placed our expression data on github.

Reviewer#2

Summary: Glioblastoma is a complex disease that exhibits heterogeneity between patients and between tumors. To date, the standard of care treatment for glioblastoma is temozolomide. The varying response of temozolomide shows the need to develop new therapeutic targets and their associated compounds for glioblastoma treatment. Glioblastoma is propagated by the expression of pro-tumor growth factors (ie. EGFR) or deletion of tumor suppression genes (ie. PTEN, p53, Rb), the authors’ novelty utilizes chromatin modifying agents to identify new treatment options in glioblastoma. To identify viable chromatin modifying/epigenetic compounds as treatment options for glioblastoma, the authors screened 106 chromatin modifying/epigenetic compounds targeting 36 different genes first in independently derived gliomaspheres, lead compounds for each of the 12 compound classes underwent a secondary screening using early passage of additional twelve gliomasphere lines. The authors calculated “delta AUC” to identify lead compound for each compound class. Of all compounds tested, CPI203, a BET class inhibitor reduced cell viability the most. RNA-seq of compound treated vs nontreated cells revealed that the same compound showed varying response in different cell line. However, CPI203, a BET class inhibitor shows the most consisted gene disruption in different cell lines. Further, the authors performed weighted correlation gene net analysis (WCGNA) ot identify sets of co-varying genes after compound treatment because genes often function through a network. WCGNA identified enriched genes “MELK” and “FBXO5” in sensitive cells, both of which as involved in mitotic and cell cycle, suggesting these two genes may serve as new therapeutic targets for glioblastoma treatment. Lastly, the author performed ATAC-seq in glioblastoma cell line after treatment with lead BET inhibitor(CPI203, JQ1, and OTX015). ATAC-seq showed a reduction of gene accessibility in cell lines after BET inhibitor treatment suggesting specificity of lead compounds.

Comments

1. In panel 1. The authors performed 106 compound screening in 5 cell lines. Each cell line was treated with 1uM or 10 uM for three days and 500 nM for twenty-eight days. Lead compound was identified by calculating “delta AUC” by the difference of day three and day twenty-eight by MTT assay.

a. In the twenty-eight day treatment, were the cells passaged as they become confluent? If cells were passaged, then the conditions of the twenty-eight-day treatment would be different compared to that of three-day treatment. If the goal was to identify lead compound by three days in each cell lines, why not use a vehicle control such as DMSO or solvent used to dilute each compound, then compare cell viability at day three to the vehicle control.

-This is an excellent point and highlights one of the main challenges of creating comparable datasets across different drugs and cell lines. In this case to avoid the problems associated with passaging of cells, we did not passage the cells. Cell numbers were calculated such that they would not overgrow the well in during this period.

The purpose of the three-day time point was not to identify lead compounds. It was actually the opposite. The purpose of the three-day time point is to eliminate compounds that killed cells too quickly and were suspected not to be working through chromatin modifications but rather “non-specific” toxicity. The authors assumed that compounds working through chromatin alteration and gene expression would take longer than three days. As such the identified compounds had big effects at 28 days and minimal effects at three days. In each of these time points the cell number was compared to a vehicle/control (e.g. DMSO). The text has been amended to clarify this point.

Minor Comments

1. In figure 2F, the authors performed RNA-seq in compound treated vs control of each cell lines. PCA clustering analysis revealed two cluster of sensitive and resistant cells. Cluster 1, the sensitive cluster seems to have varying transcriptome across PCA1 whereas the cells in cluster 2 are more similar. Gene expression of cluster 2 may reveal additional gene sets in cluster 2 that can explain drug resistance.

a. Cluster 1 was observed to express higher well-known tumor suppressor genes (CHKN2A/CDKN2B), could CPI203 serve as conditional therapy?

-The authors are thinking along similar lines. The PCA analysis revealed two clusters that showed a difference in sensitivity to multiple compounds. One cluster was generally resistant to multiple compounds. The other cluster was more sensitive to multiple compounds. Rather than recommending CPI203 treatment for this more generally sensitive cluster (high expression CDKN2A/B), we are recommending that further research and clinical trials be structured to include such refractory specimens for which current therapies may not be effective.

b. Cluster 1 seems to have varied expression across PCA1. Could this suggest that CPI203 is binding to a specific target rather than a transcriptome similarity?

-It is believed that the CPI203 compound exerts its influence by binding to the BRD proteins of BET proteins (e.g. BRD2, BRD3, BRD4 and BRDT). These BET proteins affect gene expression by binding to acetylated lysine residues on histones. The authors were surprised to see that the genetic effects were so different from cell line to cell line. However, we still believe these drugs are working through BET proteins. The fact that one cluster in the PCA analysis was more sensitive than the other probably has more to do the relative fragility or robustness of the cell lines than actually hitting a different target.

2. The authors made mentioned that the compounds used in the study were extensive but not exhaustive. Perhaps the authors should include rationale of why these compounds or specifically 106 was chosen.

-The drugs were selected as part of an epigenetic modifying drug screen set available via Tocris. The authors decided it would be helpful to have a single vendor provide all the drugs in the screen to provide more uniform quality and preparation.

---

## [Editor Report · Decision Letter 1]

16 Mar 2025

CPI203, a BET inhibitor, down-regulates a consistent set of DNA synthesis genes across a wide array of glioblastoma lines.

PONE-D-24-24126R1

Dear Dr. Garrett,

We’re pleased to inform you that your manuscript has been judged scientifically suitable for publication and will be formally accepted for publication once it meets all outstanding technical requirements.

Kind regards,

Mingli Li

Academic Editor

PLOS ONE

---

## [Editor Report · Acceptance letter]

PONE-D-24-24126R1

PLOS ONE

Dear Dr. Garrett,

I'm pleased to inform you that your manuscript has been deemed suitable for publication in PLOS ONE. Congratulations! Your manuscript is now being handed over to our production team.

Kind regards,

on behalf of

Dr. Mingli Li

Academic Editor

PLOS ONE